# MMD GAN: Towards Deeper Understanding of Moment Matching Network

**Chun-Liang Li**[1,*]    **Wei-Cheng Chang**[1,*]    **Yu Cheng**[2]    **Yiming Yang**[1]    **Barnabás Póczos**[1]
[1] Carnegie Mellon University,    [2] AI Foundations, IBM Research
{chunlial,wchang2,yiming,bapoczos}@cs.cmu.edu    chengyu@us.ibm.com
(* denotes equal contribution)

## Abstract

Generative moment matching network (GMMN) is a deep generative model that differs from Generative Adversarial Network (GAN) by replacing the discriminator in GAN with a two-sample test based on kernel maximum mean discrepancy (MMD). Although some theoretical guarantees of MMD have been studied, the empirical performance of GMMN is still not as competitive as that of GAN on challenging and large benchmark datasets. The computational efficiency of GMMN is also less desirable in comparison with GAN, partially due to its requirement for a rather large batch size during the training. In this paper, we propose to improve both the model expressiveness of GMMN and its computational efficiency by introducing *adversarial kernel learning* techniques, as the replacement of a fixed Gaussian kernel in the original GMMN. The new approach combines the key ideas in both GMMN and GAN, hence we name it *MMD GAN*. The new distance measure in MMD GAN is a meaningful loss that enjoys the advantage of weak* topology and can be optimized via gradient descent with relatively small batch sizes. In our evaluation on multiple benchmark datasets, including MNIST, CIFAR-10, CelebA and LSUN, the performance of MMD GAN significantly outperforms GMMN, and is competitive with other representative GAN works.

## 1 Introduction

The essence of unsupervised learning models the underlying distribution $\mathbb{P}_{\mathcal{X}}$ of the data $\mathcal{X}$. *Deep generative model* [1, 2] uses deep learning to approximate the distribution of complex datasets with promising results. However, modeling arbitrary density is a statistically challenging task [3]. In many applications, such as caption generation [4], accurate density estimation is not even necessary since we are only interested in *sampling* from the approximated distribution.

Rather than estimating the density of $\mathbb{P}_{\mathcal{X}}$, Generative Adversarial Network (GAN) [5] starts from a base distribution $\mathbb{P}_{\mathcal{Z}}$ over $\mathcal{Z}$, such as Gaussian distribution, then trains a transformation network $g_\theta$ such that $\mathbb{P}_\theta \approx \mathbb{P}_{\mathcal{X}}$, where $\mathbb{P}_\theta$ is the underlying distribution of $g_\theta(z)$ and $z \sim \mathbb{P}_{\mathcal{Z}}$. During the training, GAN-based algorithms require an auxiliary network $f_\phi$ to estimate the distance between $\mathbb{P}_{\mathcal{X}}$ and $\mathbb{P}_\theta$. Different probabilistic (pseudo) metrics have been studied [5–8] under GAN framework.

Instead of training an auxiliary network $f_\phi$ for measuring the distance between $\mathbb{P}_{\mathcal{X}}$ and $\mathbb{P}_\theta$, Generative moment matching network (GMMN) [9, 10] uses kernel maximum mean discrepancy (MMD) [11], which is the centerpiece of nonparametric two-sample test, to determine the distribution distances. During the training, $g_\theta$ is trained to pass the hypothesis test (minimize MMD distance). [11] shows even the simple Gaussian kernel enjoys the strong theoretical guarantees (Theorem 1). However, the empirical performance of GMMN does not meet its theoretical properties. There is no promising empirical results comparable with GAN on challenging benchmarks [12, 13]. Computationally,

it also requires larger batch size than GAN needs for training, which is considered to be less efficient [9, 10, 14, 8]

In this work, we try to improve GMMN and consider using MMD with adversarially learned kernels instead of fixed Gaussian kernels to have better hypothesis testing power. The main contributions of this work are:

- In Section 2, we prove that training $g_\theta$ via MMD with learned kernels is continuous and differentiable, which guarantees the model can be trained by gradient descent. Second, we prove a new distance measure via kernel learning, which is a sensitive loss function to the distance between $\mathbb{P}_\mathcal{X}$ and $\mathbb{P}_\theta$ (weak* topology). Empirically, the loss decreases when two distributions get closer.

- In Section 3, we propose a practical realization called MMD GAN that learns generator $g_\theta$ with the adversarially trained kernel. We further propose a feasible set reduction to speed up and stabilize the training of MMD GAN.

- In Section 5, we show that MMD GAN is computationally more efficient than GMMN, which can be trained with much smaller batch size. We also demonstrate that MMD GAN has promising results on challenging datasets, including CIFAR-10, CelebA and LSUN, where GMMN fails. To our best knowledge, we are the first MMD based work to achieve comparable results with other GAN works on these datasets.

Finally, we also study the connection to existing works in Section 4. Interestingly, we show Wasserstein GAN [8] is the special case of the proposed MMD GAN under certain conditions. The unified view shows more connections between moment matching and GAN, which can potentially inspire new algorithms based on well-developed tools in statistics [15]. Our experiment code is available at https://github.com/OctoberChang/MMD-GAN.

## 2   GAN, Two-Sample Test and GMMN

Assume we are given data $\{x_i\}_{i=1}^n$, where $x_i \in \mathcal{X}$ and $x_i \sim \mathbb{P}_\mathcal{X}$. If we are interested in sampling from $\mathbb{P}_\mathcal{X}$, it is not necessary to estimate the density of $\mathbb{P}_\mathcal{X}$. Instead, Generative Adversarial Network (GAN) [5] trains a generator $g_\theta$ parameterized by $\theta$ to transform samples $z \sim \mathbb{P}_\mathcal{Z}$, where $z \in \mathcal{Z}$, into $g_\theta(z) \sim \mathbb{P}_\theta$ such that $\mathbb{P}_\theta \approx \mathbb{P}_\mathcal{X}$. To measure the similarity between $\mathbb{P}_\mathcal{X}$ and $\mathbb{P}_\theta$ via their samples $\{x\}_{i=1}^n$ and $\{g_\theta(z_j)\}_{j=1}^n$ during the training, [5] trains the discriminator $f_\phi$ parameterized by $\phi$ for help. The learning is done by playing a two-player game, where $f_\phi$ tries to distinguish $x_i$ and $g_\theta(z_j)$ while $g_\theta$ aims to confuse $f_\phi$ by generating $g_\theta(z_j)$ similar to $x_i$.

On the other hand, distinguishing two distributions by finite samples is known as *Two-Sample Test* in statistics. One way to conduct two-sample test is via kernel maximum mean discrepancy (MMD) [11]. Given two distributions $\mathbb{P}$ and $\mathbb{Q}$, and a kernel $k$, the square of MMD distance is defined as

$$M_k(\mathbb{P}, \mathbb{Q}) = \|\mu_\mathbb{P} - \mu_\mathbb{Q}\|_\mathcal{H}^2 = \mathbb{E}_\mathbb{P}[k(x, x')] - 2\mathbb{E}_{\mathbb{P}, \mathbb{Q}}[k(x, y)] + \mathbb{E}_\mathbb{Q}[k(y, y')].$$

**Theorem 1.** *[11] Given a kernel $k$, if $k$ is a characteristic kernel, then $M_k(\mathbb{P}, \mathbb{Q}) = 0$ iff $\mathbb{P} = \mathbb{Q}$.*

**GMMN:** One example of characteristic kernel is Gaussian kernel $k(x, x') = \exp(\|x - x'\|^2)$. Based on Theorem 1, [9, 10] propose generative moment-matching network (GMMN), which trains $g_\theta$ by

$$\min_\theta M_k(\mathbb{P}_\mathcal{X}, \mathbb{P}_\theta), \tag{1}$$

with a fixed Gaussian kernel $k$ rather than training an additional discriminator $f$ as GAN.

### 2.1   MMD with Kernel Learning

In practice we use finite samples from distributions to estimate MMD distance. Given $X = \{x_1, \cdots, x_n\} \sim \mathbb{P}$ and $Y = \{y_1, \cdots, y_n\} \sim \mathbb{Q}$, one estimator of $M_k(\mathbb{P}, \mathbb{Q})$ is

$$\hat{M}_k(X, Y) = \frac{1}{\binom{n}{2}} \sum_{i \neq i'} k(x_i, x_i') - \frac{2}{\binom{n}{2}} \sum_{i \neq j} k(x_i, y_j) + \frac{1}{\binom{n}{2}} \sum_{j \neq j'} k(y_j, y_j').$$

Because of the sampling variance, $\hat{M}(X, Y)$ may not be zero even when $\mathbb{P} = \mathbb{Q}$. We then conduct hypothesis test with null hypothesis $H_0 : \mathbb{P} = \mathbb{Q}$. For a given allowable probability of false rejection $\alpha$,

we can only reject $H_0$, which imply $\mathbb{P} \neq \mathbb{Q}$, if $\hat{M}(X, Y) > c_\alpha$ for some chose threshold $c_\alpha > 0$. Otherwise, $\mathbb{Q}$ passes the test and $\mathbb{Q}$ is indistinguishable from $\mathbb{P}$ under this test. Please refer to [11] for more details.

Intuitively, if kernel $k$ cannot result in high MMD distance $M_k(\mathbb{P}, \mathbb{Q})$ when $\mathbb{P} \neq \mathbb{Q}$, $\hat{M}_k(\mathbb{P}, \mathbb{Q})$ has more chance to be smaller than $c_\alpha$. Then we are unlikely to reject the null hypothesis $H_0$ with finite samples, which implies $\mathbb{Q}$ is not distinguishable from $\mathbb{P}$. Therefore, instead of training $g_\theta$ via (1) with a pre-specified kernel $k$ as GMMN, we consider training $g_\theta$ via

$$\min_\theta \max_{k \in \mathcal{K}} M_k(\mathbb{P}_\mathcal{X}, \mathbb{P}_\theta), \tag{2}$$

which takes different possible characteristic kernels $k \in \mathcal{K}$ into account. On the other hand, we could also view (2) as replacing the fixed kernel $k$ in (1) with the *adversarially learned kernel* $\arg\max_{k \in \mathcal{K}} M_k(\mathbb{P}_\mathcal{X}, \mathbb{P}_\theta)$ to have stronger signal where $\mathbb{P} \neq \mathbb{P}_\theta$ to train $g_\theta$. We refer interested readers to [16] for more rigorous discussions about testing power and increasing MMD distances.

However, it is difficult to optimize over all characteristic kernels when we solve (2). By [11, 17] if $f$ is a injective function and $k$ is characteristic, then the resulted kernel $\tilde{k} = k \circ f$, where $\tilde{k}(x, x') = k(f(x), f(x'))$ is still characteristic. If we have a family of injective functions parameterized by $\phi$, which is denoted as $f_\phi$, we are able to change the objective to be

$$\min_\theta \max_\phi M_{k \circ f_\phi}(\mathbb{P}_\mathcal{X}, \mathbb{P}_\theta), \tag{3}$$

In this paper, we consider the case that combining Gaussian kernels with injective functions $f_\phi$, where $\tilde{k}(x, x') = \exp(-\|f_\phi(x) - f_\phi(x)'\|^2)$. One example function class of $f$ is $\{f_\phi | f_\phi(x) = \phi x, \phi > 0\}$, which is equivalent to the kernel bandwidth tuning. A more complicated realization will be discussed in Section 3. Next, we abuse the notation $M_{f_\phi}(\mathbb{P}, \mathbb{Q})$ to be MMD distance given the composition kernel of Gaussian kernel and $f_\phi$ in the following. Note that [18] considers the linear combination of characteristic kernels, which can also be incorporated into the discussed composition kernels. A more general kernel is studied in [19].

## 2.2 Properties of MMD with Kernel Learning

[8] discuss different distances between distributions adopted by existing deep learning algorithms, and show many of them are discontinuous, such as Jensen-Shannon divergence [5] and Total variation [7], except for Wasserstein distance. The discontinuity makes the gradient descent infeasible for training. From (3), we train $g_\theta$ via minimizing $\max_\phi M_{f_\phi}(\mathbb{P}_\mathcal{X}, \mathbb{P}_\theta)$. Next, we show $\max_\phi M_{f_\phi}(\mathbb{P}_\mathcal{X}, \mathbb{P}_\theta)$ also enjoys the advantage of being a continuous and differentiable objective in $\theta$ under mild assumptions.

**Assumption 2.** $g : \mathcal{Z} \times \mathbb{R}^m \to \mathcal{X}$ *is locally Lipschitz, where* $\mathcal{Z} \subseteq \mathbb{R}^d$. *We will denote* $g_\theta(z)$ *the evaluation on* $(z, \theta)$ *for convenience. Given* $f_\phi$ *and a probability distribution* $\mathbb{P}_z$ *over* $\mathcal{Z}$, $g$ *satisfies Assumption 2 if there are local Lipschitz constants* $L(\theta, z)$ *for* $f_\phi \circ g$, *which is independent of* $\phi$, *such that* $\mathbb{E}_{z \sim \mathbb{P}_z}[L(\theta, z)] < +\infty$.

**Theorem 3.** *The generator function* $g_\theta$ *parameterized by* $\theta$ *is under Assumption 2. Let* $\mathbb{P}_\mathcal{X}$ *be a fixed distribution over* $\mathcal{X}$ *and* $Z$ *be a random variable over the space* $\mathcal{Z}$. *We denote* $\mathbb{P}_\theta$ *the distribution of* $g_\theta(Z)$, *then* $\max_\phi M_{f_\phi}(\mathbb{P}_\mathcal{X}, \mathbb{P}_\theta)$ *is continuous everywhere and differentiable almost everywhere in* $\theta$.

If $g_\theta$ is parameterized by a feed-forward neural network, it satisfies Assumption 2 and can be trained via gradient descent as well as propagation, since the objective is continuous and differentiable followed by Theorem 3. More technical discussions are shown in Appendix B.

**Theorem 4.** *(weak\* topology) Let* $\{\mathbb{P}_n\}$ *be a sequence of distributions. Considering* $n \to \infty$, *under mild Assumption,* $\max_\phi M_{f_\phi}(\mathbb{P}_\mathcal{X}, \mathbb{P}_n) \to 0 \iff \mathbb{P}_n \xrightarrow{D} \mathbb{P}_\mathcal{X}$, *where* $\xrightarrow{D}$ *means converging in distribution [3].*

Theorem 4 shows that $\max_\phi M_{f_\phi}(\mathbb{P}_\mathcal{X}, \mathbb{P}_n)$ is a sensible cost function to the distance between $\mathbb{P}_\mathcal{X}$ and $\mathbb{P}_n$. The distance is decreasing when $\mathbb{P}_n$ is getting closer to $\mathbb{P}_\mathcal{X}$, which benefits the supervision of the improvement during the training. All proofs are omitted to Appendix A. In the next section, we introduce a practical realization of training $g_\theta$ via optimizing $\min_\theta \max_\phi M_{f_\phi}(\mathbb{P}_\mathcal{X}, \mathbb{P}_\theta)$.

# 3 MMD GAN

To approximate (3), we use neural networks to parameterized $g_\theta$ and $f_\phi$ with expressive power. For $g_\theta$, the assumption is locally Lipschitz, where commonly used feed-forward neural networks satisfy this constraint. Also, the gradient $\nabla_\theta \left( \max_\phi f_\phi \circ g_\theta \right)$ has to be bounded, which can be done by clipping $\phi$ [8] or gradient penalty [20]. The non-trivial part is $f_\phi$ has to be injective. For an injective function $f$, there exists an function $f^{-1}$ such that $f^{-1}(f(x)) = x, \forall x \in \mathcal{X}$ and $f^{-1}(f(g(z))) = g(z), \forall z \in \mathcal{Z}$[1], which can be approximated by an autoencoder. In the following, we denote $\phi = \{\phi_e, \phi_d\}$ to be the parameter of discriminator networks, which consists of an encoder $f_{\phi_e}$, and train the corresponding decoder $f_{\phi_d} \approx f^{-1}$ to regularize $f$. The objective (3) is relaxed to be

$$\min_\theta \max_\phi M_{f_{\phi_e}}(\mathbb{P}(\mathcal{X}), \mathbb{P}(g_\theta(\mathcal{Z}))) - \lambda \mathbb{E}_{y \in \mathcal{X} \cup g(\mathcal{Z})} \|y - f_{\phi_d}(f_{\phi_e}(y))\|^2. \qquad (4)$$

Note that we ignore the autoencoder objective when we train $\theta$, but we use (4) for a concise presentation. We note that the empirical study suggests autoencoder objective is not necessary to lead the successful GAN training as we will show in Section 5, even though the injective property is required in Theorem 1.

The proposed algorithm is similar to GAN [5], which aims to optimize two neural networks $g_\theta$ and $f_\phi$ in a minmax formulation, while the meaning of the objective is different. In [5], $f_{\phi_e}$ is a discriminator (binary) classifier to distinguish two distributions. In the proposed algorithm, distinguishing two distribution is still done by two-sample test via MMD, but with an adversarially learned kernel parametrized by $f_{\phi_e}$. $g_\theta$ is then trained to pass the hypothesis test. More connection and difference with related works is discussed in Section 4. Because of the similarity of GAN, we call the proposed algorithm *MMD GAN*. We present an implementation with the weight clipping in Algorithm 1, but one can easily extend to other Lipschitz approximations, such as gradient penalty [20].

---

**Algorithm 1:** MMD GAN, our proposed algorithm.

---

**input** : $\alpha$ the learning rate, $c$ the clipping parameter, $B$ the batch size, $n_c$ the number of iterations of discriminator per generator update.

initialize generator parameter $\theta$ and discriminator parameter $\phi$;

**while** $\theta$ *has not converged* **do**

    **for** $t = 1, \ldots, n_c$ **do**

        Sample a minibatches $\{x_i\}_{i=1}^B \sim \mathbb{P}(\mathcal{X})$ and $\{z_j\}_{j=1}^B \sim \mathbb{P}(\mathcal{Z})$

        $g_\phi \leftarrow \nabla_\phi M_{f_{\phi_e}}(\mathbb{P}(\mathcal{X}), \mathbb{P}(g_\theta(\mathcal{Z}))) - \lambda \mathbb{E}_{y \in \mathcal{X} \cup g(\mathcal{Z})} \|y - f_{\phi_d}(f_{\phi_e}(y))\|^2$

        $\phi \leftarrow \phi + \alpha \cdot \text{RMSProp}(\phi, g_\phi)$

        $\phi \leftarrow \text{clip}(\phi, -c, c)$

    Sample a minibatches $\{x_i\}_{i=1}^B \sim \mathbb{P}(\mathcal{X})$ and $\{z_j\}_{j=1}^B \sim \mathbb{P}(\mathcal{Z})$

    $g_\theta \leftarrow \nabla_\theta M_{f_{\phi_e}}(\mathbb{P}(\mathcal{X}), \mathbb{P}(g_\theta(\mathcal{Z})))$

    $\theta \leftarrow \theta - \alpha \cdot \text{RMSProp}(\theta, g_\theta)$

---

**Encoding Perspective of MMD GAN:** Besides from using kernel selection to explain MMD GAN, the other way to see the proposed MMD GAN is viewing $f_{\phi_e}$ as a feature transformation function, and the kernel two-sample test is performed on this transformed feature space (i.e., the code space of the autoencoder). The optimization is finding a manifold with stronger signals for MMD two-sample test. From this perspective, [9] is the special case of MMD GAN if $f_{\phi_e}$ is the identity mapping function. In such circumstance, the kernel two-sample test is conducted in the original data space.

## 3.1 Feasible Set Reduction

**Theorem 5.** *For any $f_\phi$, there exists $f'_\phi$ such that $M_{f_\phi}(\mathbb{P}_r, \mathbb{P}_\theta) = M_{f'_\phi}(\mathbb{P}_r, \mathbb{P}_\theta)$ and $\mathbb{E}_x[f_\phi(x)] \succeq \mathbb{E}_z[f_{\phi'}(g_\theta(z))]$.*

With Theorem 5, we could reduce the feasible set of $\phi$ during the optimization by solving

$$\min_\theta \max_\phi M_{f_\phi}(\mathbb{P}_r, \mathbb{P}_\theta) \quad s.t. \ \mathbb{E}[f_\phi(x)] \succeq \mathbb{E}[f_\phi(g_\theta(z))]$$

which the optimal solution is still *equivalent* to solving (2).

However, it is hard to solve the constrained optimization problem with backpropagation. We relax the constraint by ordinal regression [21] to be

$$\min_\theta \max_\phi M_{f_\phi}(\mathbb{P}_r, \mathbb{P}_\theta) + \lambda \min \left( \mathbb{E}[f_\phi(x)] - \mathbb{E}[f_\phi(g_\theta(z))], 0 \right),$$

which only penalizes the objective when the constraint is violated. In practice, we observe that reducing the feasible set makes the training faster and stabler.

## 4   Related Works

There has been a recent surge on improving GAN [5]. We review some related works here.

**Connection with WGAN:**  If we composite $f_\phi$ with linear kernel instead of Gaussian kernel, and restricting the output dimension $h$ to be 1, we then have the objective

$$\min_\theta \max_\phi \|\mathbb{E}[f_\phi(x)] - \mathbb{E}[f_\phi(g_\theta(z))]\|^2. \tag{5}$$

Parameterizing $f_\phi$ and $g_\theta$ with neural networks and assuming $\exists \phi' \in \Phi$ such $f'_\phi = -f_\phi, \forall \Phi$, recovers Wasserstein GAN (WGAN) [8] [2]. If we treat $f_\phi(x)$ as the data transform function, WGAN can be interpreted as first-order moment matching (linear kernel) while MMD GAN aims to match infinite order of moments with Gaussian kernel form Taylor expansion [9]. Theoretically, Wasserstein distance has similar theoretically guarantee as Theorem 1, 3 and 4. In practice, [22] show neural networks does not have enough capacity to approximate Wasserstein distance. In Section 5, we demonstrate matching high-order moments benefits the results. [23] also propose McGAN that matches second order moment from the primal-dual norm perspective. However, the proposed algorithm requires matrix (tensor) decompositions because of exact moment matching [24], which is hard to scale to higher order moment matching. On the other hand, by giving up exact moment matching, MMD GAN can match high-order moments with kernel tricks. More detailed discussions are in Appendix B.3.

**Difference from Other Works with Autoencoders:**   Energy-based GANs [7, 25] also utilizes the autoencoder (AE) in its discriminator from the energy model perspective, which minimizes the reconstruction error of real samples $x$ while maximize the reconstruction error of generated samples $g_\theta(z)$. In contrast, MMD GAN uses AE to approximate invertible functions by minimizing the reconstruction errors of *both* real samples $x$ and generated samples $g_\theta(z)$. Also, [8] show EBGAN approximates total variation, with the drawback of discontinuity, while MMD GAN optimizes MMD distance. The other line of works  [2, 26, 9] aims to match the AE codespace $f(x)$, and utilize the decoder $f_{dec}(\cdot)$. [2, 26] match the distribution of $f(x)$ and $z$ via different distribution distances and generate data (e.g. image) by $f_{dec}(z)$. [9] use MMD to match $f(x)$ and $g(z)$, and generate data via $f_{dec}(g(z))$. The proposed MMD GAN matches the $f(x)$ and $f(g(z))$, and generates data via $g(z)$ directly as GAN. [27] is similar to MMD GAN but it considers KL-divergence without showing continuity and weak$^*$ topology guarantee as we prove in Section 2.

**Other GAN Works:**   In addition to the discussed works, there are several extended works of GAN. [28] proposes using the linear kernel to match first moment of its discriminator's latent features. [14] considers the variance of empirical MMD score during the training. Also, [14] only improves the latent feature matching in [28] by using kernel MMD, instead of proposing an adversarial training framework as we studied in Section 2. [29] uses Wasserstein distance to match the distribution of autoencoder loss instead of data. One can consider to extend [29] to higher order matching based on the proposed MMD GAN. A parallel work [30] use energy distance, which can be treated as MMD GAN with different kernel. However, there are some potential problems of its critic. More discussion can be referred to [31].

## 5   Experiment

We train MMD GAN for image generation on the MNIST [32], CIFAR-10 [33], CelebA [13], and LSUN bedrooms [12] datasets, where the size of training instances are 50K, 50K, 160K, 3M

respectively. All the samples images are generated from a fixed noise random vectors and are not cherry-picked.

**Network architecture:** In our experiments, we follow the architecture of DCGAN [34] to design $g_\theta$ by its generator and $f_\phi$ by its discriminator except for expanding the output layer of $f_\phi$ to be $h$ dimensions.

**Kernel designs:** The loss function of MMD GAN is implicitly associated with a family of characteristic kernels. Similar to the prior MMD seminal papers [10, 9, 14], we consider a mixture of $K$ RBF kernels $k(x, x') = \sum_{q=1}^{K} k_{\sigma_q}(x, x')$ where $k_{\sigma_q}$ is a Gaussian kernel with bandwidth parameter $\sigma_q$. Tuning kernel bandwidth $\sigma_q$ optimally still remains an open problem. In this works, we fixed $K = 5$ and $\sigma_q$ to be $\{1, 2, 4, 8, 16\}$ and left the $f_\phi$ to learn the kernel (feature representation) under these $\sigma_q$.

**Hyper-parameters:** We use RMSProp [35] with learning rate of 0.00005 for a fair comparison with WGAN as suggested in its original paper [8]. We ensure the boundedness of model parameters of discriminator by clipping the weights point-wisely to the range $[-0.01, 0.01]$ as required by Assumption 2. The dimensionality $h$ of the latent space is manually set according to the complexity of the dataset. We thus use $h = 16$ for MNIST, $h = 64$ for CelebA, and $h = 128$ for CIFAR-10 and LSUN bedrooms. The batch size is set to be $B = 64$ for all datasets.

## 5.1 Qualitative Analysis

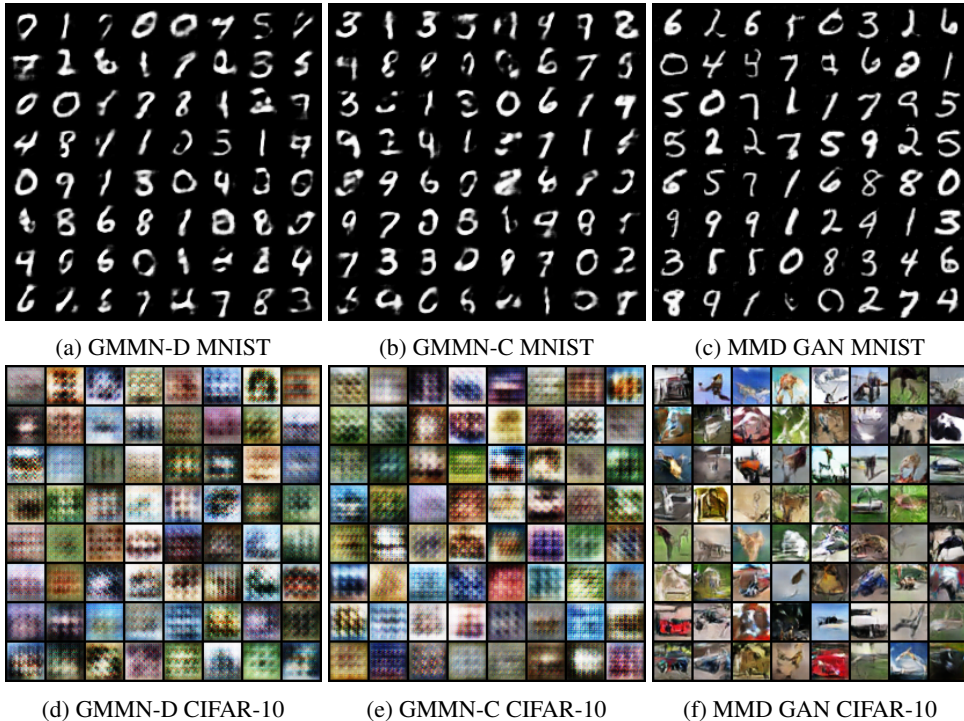

(a) GMMN-D MNIST     (b) GMMN-C MNIST     (c) MMD GAN MNIST

(d) GMMN-D CIFAR-10     (e) GMMN-C CIFAR-10     (f) MMD GAN CIFAR-10

Figure 1: Generated samples from GMMN-D (Dataspace), GMMN-C (Codespace) and our MMD GAN with batch size $B = 64$.

We start with comparing MMD GAN with GMMN on two standard benchmarks, MNIST and CIFAR-10. We consider two variants for GMMN. The first one is original GMMN, which trains the generator by minimizing the MMD distance on the original data space. We call it as *GMMN-D*. To compare with MMD GAN, we also pretrain an autoencoder for projecting data to a manifold, then fix the autoencoder as a feature transformation, and train the generator by minimizing the MMD distance in the code space. We call it as *GMMN-C*.

The results are pictured in Figure 1. Both GMMN-D and GMMN-C are able to generate meaningful digits on MNIST because of the simple data structure. By a closer look, nonetheless, the boundary and shape of the digits in Figure 1a and 1b are often irregular and non-smooth. In contrast, the sample

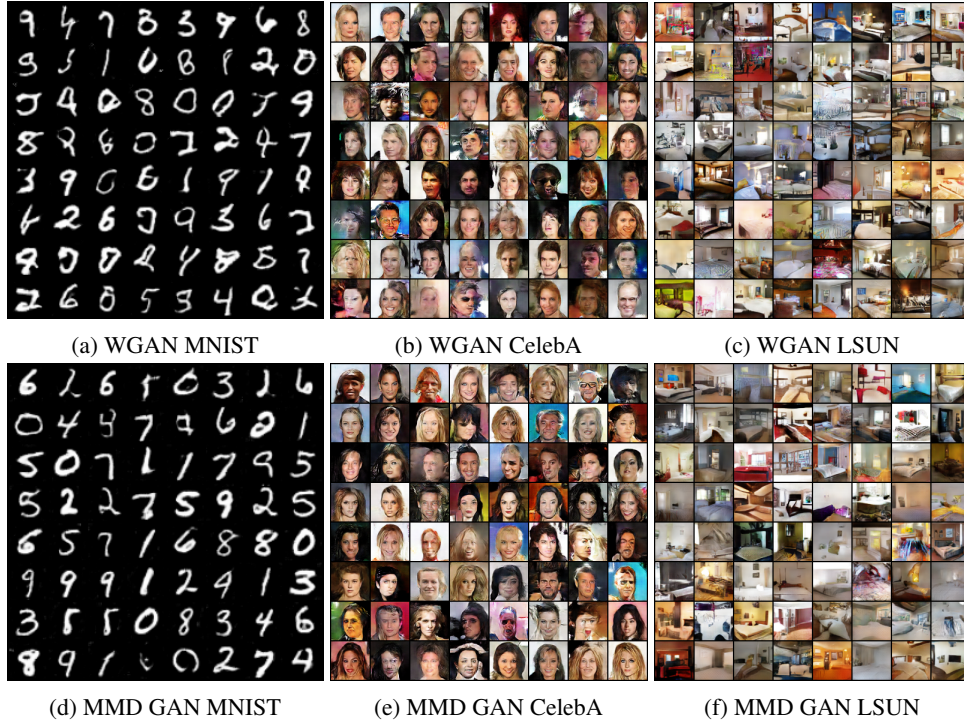

| (a) WGAN MNIST | (b) WGAN CelebA | (c) WGAN LSUN |
| (d) MMD GAN MNIST | (e) MMD GAN CelebA | (f) MMD GAN LSUN |

Figure 2: Generated samples from WGAN and MMD GAN on MNIST, CelebA, and LSUN bedroom datasets.

digits in Figure 1c are more natural with smooth outline and sharper strike. For CIFAR-10 dataset, both GMMN variants fail to generate meaningful images, but resulting some low level visual features. We observe similar cases in other complex large-scale datasets such as CelebA and LSUN bedrooms, thus results are omitted. On the other hand, the proposed MMD GAN successfully outputs natural images with sharp boundary and high diversity. The results in Figure 1 confirm the success of the proposed adversarial learned kernels to enrich statistical testing power, which is the key difference between GMMN and MMD GAN.

If we increase the batch size of GMMN to $1024$, the image quality is improved, however, it is still not competitive to MMD GAN with $B = 64$. The images are put in Appendix C. This demonstrates that the proposed MMD GAN can be trained more efficiently than GMMN with smaller batch size.

**Comparisons with GANs:** There are several representative extensions of GANs. We consider recent state-of-art WGAN [8] based on DCGAN structure [34], because of the connection with MMD GAN discussed in Section 4. The results are shown in Figure 2. For MNIST, the digits generated from WGAN in Figure 2a are more unnatural with peculiar strikes. In Contrary, the digits from MMD GAN in Figure 2d enjoy smoother contour. Furthermore, both WGAN and MMD GAN generate diversified digits, avoiding the mode collapse problems appeared in the literature of training GANs. For CelebA, we can see the difference of generated samples from WGAN and MMD GAN. Specifically, we observe varied poses, expressions, genders, skin colors and light exposure in Figure 2b and 2e. By a closer look (view on-screen with zooming in), we observe that faces from WGAN have higher chances to be blurry and twisted while faces from MMD GAN are more spontaneous with sharp and acute outline of faces. As for LSUN dataset, we could not distinguish salient differences between the samples generated from MMD GAN and WGAN.

## 5.2 Quantitative Analysis

To quantitatively measure the quality and diversity of generated samples, we compute the inception score [28] on CIFAR-10 images. The inception score is used for GANs to measure samples quality and diversity on the pretrained inception model [28]. Models that generate collapsed samples have a relatively low score. Table 1 lists the results for $50K$ samples generated by various unsupervised

generative models trained on CIFAR-10 dataset. The inception scores of [36, 37, 28] are directly derived from the corresponding references.

Although both WGAN and MMD GAN can generate sharp images as we show in Section 5.1, our score is better than other GAN techniques except for DFM [36]. This seems to confirm empirically that higher order of moment matching between the real data and fake sample distribution benefits generating more diversified sample images. Also note DFM appears compatible with our method and combing training techniques in DFM is a possible avenue for future work.

| Method | Scores $\pm$ std. |
|---|---|
| Real data | $11.95 \pm .20$ |
| DFM [36] | 7.72 |
| ALI [37] | 5.34 |
| Improved GANs [28] | 4.36 |
| MMD GAN | $\mathbf{6.17} \pm .07$ |
| WGAN | $5.88 \pm .07$ |
| GMMN-C | $3.94 \pm .04$ |
| GMMN-D | $3.47 \pm .03$ |

Table 1: Inception scores

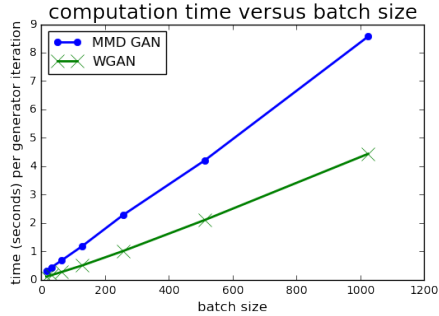

Figure 3: Computation time

## 5.3 Stability of MMD GAN

We further illustrate how the MMD distance correlates well with the quality of the generated samples. Figure 4 plots the evolution of the MMD GAN estimate the MMD distance during training for MNIST, CelebA and LSUN datasets. We report the average of the $\hat{M}_{f_\phi}(\mathbb{P}_\mathcal{X}, \mathbb{P}_\theta)$ with moving average to smooth the graph to reduce the variance caused by mini-batch stochastic training. We observe during the whole training process, samples generated from the same noise vector across iterations, remain similar in nature. (e.g., face identity and bedroom style are alike while details and backgrounds will evolve.) This qualitative observation indicates valuable stability of the training process. The decreasing curve with the improving quality of images supports the weak* topology shown in Theorem 4. Also, We can see from the plot that the model converges very quickly. In Figure 4b, for example, it converges shortly after tens of thousands of generator iterations on CelebA dataset.

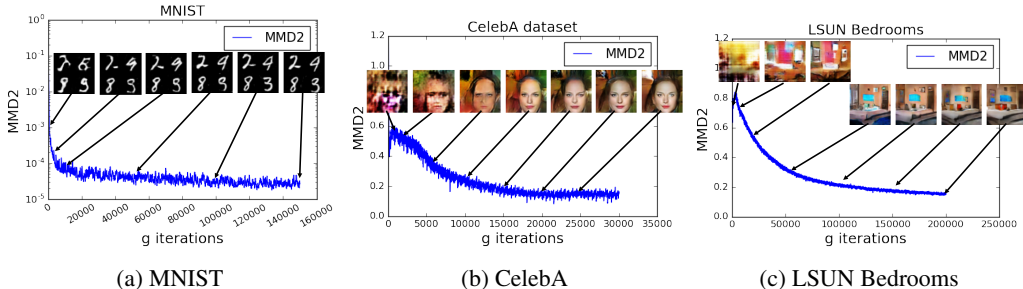

(a) MNIST      (b) CelebA      (c) LSUN Bedrooms

Figure 4: Training curves and generative samples at different stages of training. We can see a clear correlation between lower distance and better sample quality.

## 5.4 Computation Issue

We conduct time complexity analysis with respect to the batch size $B$. The time complexity of each iteration is $O(B)$ for WGAN and $O(KB^2)$ for our proposed MMD GAN with a mixture of $K$ RBF kernels. The quadratic complexity $O(B^2)$ of MMD GAN is introduced by computing kernel matrix, which is sometimes criticized for being inapplicable with large batch size in practice. However, we point that there are several recent works, such as EBGAN [7], also matching pairwise relation between samples of batch size, leading to $O(B^2)$ complexity as well.

Empirically, we find that under GPU environment, the highly parallelized matrix operation tremendously alleviated the quadratic time to almost linear time with modest $B$. Figure 3 compares the computational time per generator iterations versus different $B$ on Titan X. When $B = 64$, which is adapted for training MMD GAN in our experiments setting, the time per iteration of WGAN and MMD GAN is 0.268 and 0.676 seconds, respectively. When $B = 1024$, which is used for training GMMN in its references [9], the time per iteration becomes 4.431 and 8.565 seconds, respectively. This result coheres our argument that the empirical computational time for MMD GAN is not quadratically expensive compared to WGAN with powerful GPU parallel computation.

## 5.5 Better Lipschitz Approximation and Necessity of Auto-Encoder

Although we used weight-clipping for Lipschitz constraint in Assumption 2, one can also use other approximations, such as gradient penalty [20]. On the other hand, in Algorithm 1, we present an algorithm with auto-encoder to be consistent with the theory that requires $f_\phi$ to be injective. However, we observe that it is not necessary in practice. We show some preliminary results of training MMD GAN with gradient penalty and without the auto-encoder in Figure 5. The preliminary study indicates that MMD GAN can generate satisfactory results with other Lipschitz constraint approximation. One potential future work is conducting more thorough empirical comparison studies between different approximations.

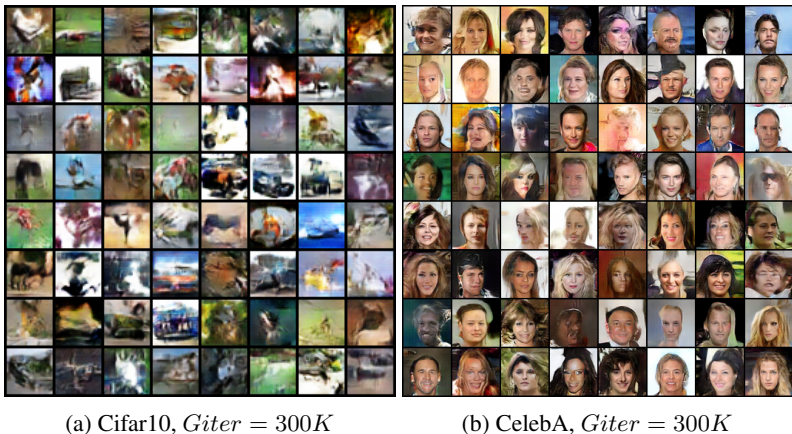

(a) Cifar10, $Giter = 300K$        (b) CelebA, $Giter = 300K$

Figure 5: MMD GAN results using gradient penalty [20] and without auto-encoder reconstruction loss during training.

## 6 Discussion

We introduce a new deep generative model trained via MMD with adversarially learned kernels. We further study its theoretical properties and propose a practical realization MMD GAN, which can be trained with much smaller batch size than GMMN and has competitive performances with state-of-the-art GANs. We can view MMD GAN as the first practical step forward connecting moment matching network and GAN. One important direction is applying developed tools in moment matching [15] on general GAN works based the connections shown by MMD GAN. Also, in Section 4, we connect WGAN and MMD GAN by first-order and infinite-order moment matching. [24] shows finite-order moment matching ($\sim 5$) achieves the best performance on domain adaption. One could extend MMD GAN to this by using polynomial kernels. Last, in theory, an injective mapping $f_\phi$ is *necessary* for the theoretical guarantees. However, we observe that it is not mandatory in practice as we show in Section 5.5. One conjecture is it usually learns the injective mapping with high probability by parameterizing with neural networks, which worth more study as a future work.

## Acknowledgments

We thank the reviewers for their helpful comments. This work is supported in part by the National Science Foundation (NSF) under grants IIS-1546329 and IIS-1563887.

## Footnotes

[1]Note that injective is not necessary invertible.

[2]Theoretically, they are not equivalent but the practical neural network approximation results in the same algorithm.

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
