[Supplementary Material]

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

# A Technical Proof

## A.1 Proof of Theorem 3

*Proof.* Since MMD is a probabilistic metric [11], we have the triangular inequality for every $M_f$. Therefore,

$$|\max_\phi M_{f_\phi}(\mathbb{P}_\mathcal{X}, \mathbb{P}_\theta) - \max_\phi M_{f_\phi}(\mathbb{P}_\mathcal{X}, \mathbb{P}_{\theta'})| \leq \max_\phi |M_{f_\phi}(\mathbb{P}_\mathcal{X}, \mathbb{P}_\theta) - M_{f_\phi}(\mathbb{P}_\mathcal{X}, \mathbb{P}_{\theta'})| \quad (6)$$

$$= |M_{f_{\phi^*}}(\mathbb{P}_\mathcal{X}, \mathbb{P}_\theta) - M_{f_{\phi^*}}(\mathbb{P}_\mathcal{X}, \mathbb{P}_{\theta'})|$$

$$\leq M_{f_{\phi^*}}(\mathbb{P}_\theta, \mathbb{P}_{\theta'}), \quad (7)$$

where $\phi^*$ is the solution of (6).

By definition, given any $\phi \in \Phi$, define $h_\theta = f_\phi \circ g_\theta$, the MMD distance $M_{f_\phi}(\mathbb{P}_\theta, \mathbb{P}_{\theta'})$

$$= \mathbb{E}_{z,z'}\left[k(h_\theta(z), h_\theta(z')) - 2k(h_\theta(z), h_{\theta'}(z')) + k(h_{\theta'}(z), h_{\theta'}(z'))\right]$$
$$\leq \mathbb{E}_{z,z'}\left[|k(h_\theta(z), h_\theta(z')) - k(h_\theta(z), h_{\theta'}(z'))|\right] + \mathbb{E}_{z,z'}\left[|k(h_{\theta'}(z), h_{\theta'}(z')) - k(h_\theta(z), h_{\theta'}(z'))|\right] \quad (8)$$

In this, we consider Gaussian kernel $k$, therefore

$$|k(h_\theta(z), h_\theta(z')) - k(h_\theta(z), h_{\theta'}(z'))| = |\exp(-\|h_\theta(z) - h_\theta(z')\|^2) - \exp(-\|h_\theta(z) - h_{\theta'}(z')\|^2)| \leq 1,$$

for all $(\theta, \theta')$ and $(z, z')$. Similarly, $|k(h_{\theta'}(z), h_{\theta'}(z')) - k(h_\theta(z), h_{\theta'}(z'))| \leq 1$. Combining the above claim with (7) and bounded convergence theorem, we have

$$|\max_\phi M_{f_\phi}(\mathbb{P}_\mathcal{X}, \mathbb{P}_\theta) - \max_\phi M_{f_\phi}(\mathbb{P}_\mathcal{X}, \mathbb{P}_{\theta'})| \xrightarrow{\theta \to \theta'} 0,$$

which proves the continuity of $\max_f MMD_f(\mathbb{P}_\mathcal{X}, \mathbb{P}_\theta)$.

By Mean Value Theorem, $|e^{-x^2} - e^{-y^2}| \leq \max_z |2ze^{-z^2}| \times |x - y| \leq |x - y|$. Incorporating it with (8) and triangular inequality, we have

$$M_{f_\phi}(\mathbb{P}_\theta, \mathbb{P}_{\theta'}) \leq \mathbb{E}_{z,z'}[\|h_\theta(z') - h_{\theta'}(z')\| + \|h_\theta(z) - h_{\theta'}(z)\|]$$
$$= 2\mathbb{E}_z[\|h_\theta(z) - h_{\theta'}(z)\|]$$

Now let $h$ be locally Lipschitz. For a given pair $(\theta, z)$ there is a constant $L(\theta, z)$ and an open set $U_\theta$ such that for every $(\theta', z) \in U_\theta$ we have

$$\|h_\theta(z) - h_{\theta'}(z)\| \leq L(\theta, z)(\|\theta - \theta'\|)$$

Under Assumption 2, we then achieve

$$|M_{f_\phi}(\mathbb{P}_\mathcal{X}, \mathbb{P}_\theta) - M_{f_\phi}(\mathbb{P}_\mathcal{X}, \mathbb{P}'_\theta)| \leq M_{f_\phi}(\mathbb{P}_\theta, \mathbb{P}_{\theta'}) \leq 2\mathbb{E}_z[L(\theta, z)]\|\theta - \theta'\|. \quad (9)$$

Combining (7) and (9) implies $\max_\phi M_{f_\phi}(\mathbb{P}_\mathcal{X}, \mathbb{P}_\theta)$ is locally Lipschitz and continuous everywhere. Last, applying Radamacher's theorem proves $\max_\phi M_{f_\phi}(\mathbb{P}_\mathcal{X}, \mathbb{P}_\theta)$ is differentiable almost everywhere, which completes the proof. □

## A.2 Proof of Theorem 4

*Proof.* The proof utilizes parts of results from [38].

($\Rightarrow$)   If $\max_\phi M_{f_\phi}(\mathbb{P}_n, \mathbb{P}) \to 0$, there exists $\phi \in \Phi$ such that $M_{f_\phi}(\mathbb{P}, \mathbb{P}_n) \to 0$ since $M_{f_\phi}(\mathbb{P}, \mathbb{P}_n)$ is non-negative. By [38], for any characteristic kernel $k$,

$$M_k(\mathbb{P}_n, \mathbb{P}) \to 0 \iff \mathbb{P}_n \xrightarrow{D} \mathbb{P}.$$

Therefore, if $\max_\phi M_{f_\phi}(\mathbb{P}_n, \mathbb{P}) \to 0$, $\mathbb{P}_n \xrightarrow{D} \mathbb{P}$.

($\Leftarrow$)   By [38], given a characteristic kernel $k$, if $\sup_{x,x'} k(x, x') \leq 1$, $\sqrt{M_k(\mathbb{P}, \mathbb{Q})} \leq W(\mathbb{P}, \mathbb{Q})$, where $W(\mathbb{P}, \mathbb{Q})$ is Wasserstein distance. In this paper, we consider the kernel $k(x, x') = \exp(-\|f_\phi(x) - f_\phi(x')\|^2) \leq 1$. By above, $\sqrt{M_{f_\phi}(\mathbb{P}, \mathbb{Q})} \leq W(\mathbb{P}, \mathbb{Q}), \forall \phi$. By [8], if $\mathbb{P}_n \xrightarrow{D} \mathbb{P}$, $W(\mathbb{P}, \mathbb{P}_n) \to 0$. Combining all of them, we get

$$\mathbb{P}_n \xrightarrow{D} \mathbb{P} \implies W(\mathbb{P}, \mathbb{P}_n) \to 0 \implies \max_\phi M_{f_\phi}(\mathbb{P}, \mathbb{P}_n) \to 0.$$

□

### A.3 Proof of Theorem 5

*Proof.* The proof assumes $f_\phi(x)$ is scalar, but the vector case can be proved with the same sketch. First, if $\mathbb{E}[f_\phi(x)] > \mathbb{E}[f_\phi(g_\theta(z))]$, then $\phi = \phi'$. If $\mathbb{E}[f_\phi(x)] < \mathbb{E}[f_\phi(g_\theta(z))]$, we let $f = -f_\phi$, then $\mathbb{E}[f(x)] > \mathbb{E}[f(g_\theta(z))]$ and flipping sign does not change the MMD distance. If we parameterized $f_\phi$ by a neural network, which has a linear output layer, $\phi'$ can realized by flipping the sign of the weights of the last layer. $\qquad\square$

## B Property of MMD with Fixed and Learned Kernels

### B.1 Continuity and Differentiability

One can simplify Theorem 3 and its proof for standard MMD distance to show MMD is also continuous and differentiable almost everywhere. In [8], they propose a counterexample to show the discontinuity of MMD by assuming $\mathcal{H} = L^2$. However, it is known that $L^2$ is not in RKHS, so the discussed counterexample is not appropriate.

### B.2 IPM Framework

From integral probability metrics (IPM), the probabilistic distance can be defined as

$$\Delta(\mathbb{P}, \mathbb{Q}) = \sup_{f \in \mathcal{F}} \mathbb{E}_{x \sim \mathbb{P}}[f(x)] - \mathbb{E}_{y \sim \mathbb{Q}}[f(y)]. \tag{10}$$

By changing the function class $\mathcal{F}$, we can recover several distances, such as total variation, Wasserstein distance and MMD distance. From [8], the discriminator $f_\phi$ in different existing works of GAN can be explained to be used to solve different probabilistic metrics based on (10). For MMD, the function class $\mathcal{F}$ is $\{\|f\|_{\mathcal{H}_k} \le 1\}$, where $\mathcal{H}$ is RKHS associated with kernel $k$. Different form many distances, such as total variation and Wasserstein distance, there is an analytical representation [11] as we show in Section 2, which is

$$\Delta(\mathbb{P}, \mathbb{Q}) = MMD(\mathbb{P}, \mathbb{Q}) = \|\mu_\mathbb{P} - \mu_\mathbb{Q}\|_\mathcal{H} = \sqrt{\mathbb{E}_\mathbb{P}[k(x, x')] - 2\mathbb{E}_{\mathbb{P},\mathbb{Q}}[k(x, y)] + \mathbb{E}_\mathbb{Q}[k(y, y')]}.$$

Because of the analytical representation of (10), GMMN does not need an additional network $f_\phi$ for estimating the distance.

Here we also provide an explanation of the proposed MMD with adversarially learned kernel under IPM framework. The MMD distance with adversarially learned kernel is represented as

$$\max_{k \in \mathcal{K}} MMD(\mathbb{P}, \mathbb{Q}),$$

The corresponding IPM formulation is

$$\Delta(\mathbb{P}, \mathbb{Q}) = \max_{k \in \mathcal{K}} MMD(\mathbb{P}, \mathbb{Q}) = \sup_{f \in \mathcal{H}_{k_1} \cup \cdots \cup \mathcal{H}_{k_n}} \mathbb{E}_{x \sim \mathbb{P}}[f(x)] - \mathbb{E}_{y \sim \mathbb{Q}}[f(y)],$$

where $k_i \in \mathcal{K}, \forall i$. From this perspective, the proposed MMD distance with adversarially learned kernel is still defined by IPM but with a larger function class.

### B.3 MMD is an Efficient Moment Matching

We consider the example of matching first and second moments of $\mathbb{P}$ and $\mathbb{Q}$. The $\ell_1$ objective in McGAN [23] is

$$\|\mu_\mathbb{P} - \mu_\mathbb{Q}\|_1 + \|\Sigma_\mathbb{P} - \Sigma_\mathbb{Q}\|_*,$$

where $\|\cdot\|_*$ is the trace norm. The objective can be changed to be general $\ell_q$ norm.

In MMD, with polynomial kernel $k(x, y) = 1 + x^\top y$, the MMD distance is

$$2\|\mu_\mathbb{P} - \mu_\mathbb{Q}\|^2 + \|\Sigma_\mathbb{P} - \Sigma_\mathbb{Q} + \mu_\mathbb{P}\mu_\mathbb{P}^\top - \mu_\mathbb{Q}\mu_\mathbb{Q}^\top\|_F^2,$$

which is *inexact* moment matching because the second term contains the quadratic of the first moment. It is difficult to match high-order moments, because we have to deal with high order tensors directly. On the other hand, MMD can easily match high-order moments (even infinite order moments by using Gaussian kernel) with kernel tricks, and enjoys strong theoretical guarantee.

# C Performance of GMMN

(a) GMMN-D on MNIST

(b) GMMN-C on MNIST

(c) GMMN-D on CIFAR-10

(d) GMMN-C CIFAR-10

Figure 6: Generative samples from GMMN-D and GMMN-C with training batch size $B = 1024$.