[Reviews · NeurIPS 2017]

Reviewer 1



This paper presents the MMD-GAN, a generative architecture which combines the benefits of generative adversarial nets (GANs) and generative moment matching nets (GMMNs). The architecture is based on MMD, but with a kernel function learned by an adversary trying to maximize MMD. There's a bit of theory showing non-degeneracy, and experiments demonstrate that the generated samples are competitive with W-GANs. Overall, I think this is a strong paper which presents a simple idea and executes it well. The proposed method is intuitively appealing. The modeling choices are well motivated, the writing is clear, the theoretical justification is nice, and relationships to other methods are clearly highlighted. The experiments compare against strong baselines on challenging datasets. Being able to make GMMNs competitive with WGANs is impressive, since to the best of my knowledge there was still a large performance gap between them. While the results seem significantly better than the baselines, the paper could use more analysis of where this improvement comes from. How does the MMD-GAN eliminate the need for large mini-batches? (Don't the samples in each mini-batch still need to essentially tile the space in order to achieve low MMD?) Is there an explanation for why it outperforms the WGAN? Minor comments: - Section 3 describes the method in relation to the GAN, but isn't the WGAN a more direct match? If I understand right, the MMD-GAN is essentially a kernelized W-GAN. - "Also, the gradient has to be bounded, which can be done by clipping phi." Is this the same as the Lipschitz constraint from WGANs? If so, why not use the regularization method from the "improved WGAN" paper, rather than the original version (which the authors found was less effective)? - If the claim is that the proposed method prevents mode dropping, estimating log-likelihoods is probably a stronger quantitative way to test it. E.g., see Wu et al., 2017. - The experiment of Section 5.3 doesn't seem to be measuring stability. (I assume this refers to avoiding the degenerate solutions that regular GANs fall into?)

Reviewer 2



This paper proposes to improve the performance of the generative moment matching network (GMMN) by learning the kernel of the MMD. The optimization will result in a min-max game similar to that of the generative adversarial networks: the adversarial kernel is trained to maximize the MMD distance of the model distribution and the data distribution, and the generator at the same time is trained to fool the MMD. The kernel of MMD needs to be characteristic and is composed of Gaussian kernels with an injective function that is learned using an autoencoder. I have some questions from the authors: The decoder of the autoencoder f_dec does not seem to be learned in Algorithm 1. Is it just the transpose of the encoder? What is the architecture of the decoder and what kind of up-sampling does it have? The encoder of the autoencoder f_\phi is the same as the discriminator of DCGAN, right? Given that the DCGAN discriminator has max-pooling layers, how does f_\phi become injective? How is the reconstruction quality of the autoencoder given that only h~100 dimensional vector is used for the latent space and given that there is max-pooling? Is it really necessary to train the autoencoder to get the model working? (this algorithm is very similar to W-GAN and there is no autoencoder there --- although I understand the autoencoder training is necessary to have an injective function.) The final algorithm can be considered as the generative moment matching in the code space of an autoencoder, where the features of the autoencoder is also trained to maximize the MMD distance. The algorithm is also very similar to WGAN where instead of only matching the means, it is matching high-order moments due to the kernel trick. So in general I wouldn't expect this algorithm to work much better that other GAN variants, however I found the intuition behind the algorithm and its connection to GMMN and W-GAN interesting.

Reviewer 3



GANs have received a lot of attention lately. They generate sharp samples but this performance comes with the cost of optimization difficulties. An alternative to GANs that relies on two-sample hypothesis testing is the so called Generative Moment Matching Networks (GMMN). Although GMMNs are more stable, they generate worse samples than GANs. They also require a large batch size during training. This paper proposes to improve the efficiency of the GMMN and its architecture for better performance. The approach consists in using feature matching in the kernel definition and learning the parameters of this feature map jointly with the parameters of the generator in a minimax problem. I liked the paper and have highlighted some remarks below: 1. The reason why adversarial learning of the kernel leads to higher test power was not thoroughly discussed. This is important. 2. The experiment section -- including the complexity analysis -- is very well carried through. 3. The paper can benefit from a thorough editing. There were many typos that tend to distract from the message being portrayed in the paper. REBUTTAL: I acknowledge the rebuttal